# TALK THE WALK: NAVIGATING GRIDS IN NEW YORK CITY THROUGH GROUNDED DIALOGUE

## ABSTRACT

We introduce "Talk The Walk", the first large-scale dialogue dataset grounded in action and perception. The task involves two agents (a "guide" and a "tourist") that communicate via natural language in order to achieve a common goal: having the tourist navigate to a given target location. The task and dataset, which are described in detail, are challenging and their full solution is an open problem that we pose to the community. We (i) focus on the task of tourist localization and develop the novel Masked Attention for Spatial Convolutions (MASC) mechanism that allows for grounding tourist utterances into the guide's map, (ii) show it yields significant improvements for both emergent and natural language communication, and (iii) using this method, we establish non-trivial baselines on the full task.

## 1 INTRODUCTION

As artificial intelligence plays an ever more prominent role in everyday human lives, it becomes increasingly important to enable machines to communicate via natural language—not only with humans, but also with each other. Learning algorithms for natural language understanding, such as in machine translation and reading comprehension, have progressed at an unprecedented rate in recent years, but still rely on static, large-scale, text-only datasets that lack crucial aspects of how humans understand and produce natural language. Namely, humans develop language capabilities by being embodied in an environment which they can perceive, manipulate and move around in; and by interacting with other humans. Hence, we argue that we should incorporate all three fundamental aspects of human language acquisition—perception, action and interactive communication—and develop a task and dataset to that effect.

We introduce the Talk the Walk dataset, where the aim is for two agents, a "guide" and a "tourist", to interact with each other via natural language in order to achieve a common goal: having the tourist navigate towards the correct location. The guide has access to a map and knows the target location, but does not know where the tourist is; the tourist has a 360-degree view of the world, but knows neither the target location on the map nor the way to it. The agents need to work together through communication in order to successfully solve the task. An example of the task is given in Figure 1.

Grounded language learning has (re-)gained traction in the AI community, and much attention is currently devoted to *virtual embodiment*—the development of multi-agent communication tasks in virtual environments—which has been argued to be a viable strategy for acquiring natural language semantics Kiela et al. (2016). Various related tasks have recently been introduced, but in each case with some limitations. Although visually grounded dialogue tasks de Vries et al. (2016); Das et al. (2016) comprise perceptual grounding and multi-agent interaction, their agents are passive observers and do not act in the environment. By contrast, instruction-following tasks, such as VNL Anderson et al. (2017), involve action and perception but lack natural language interaction with other agents. Furthermore, some of these works use simulated environments Das et al. (2017a) and/or templated language Hermann et al. (2017), which arguably oversimplifies real perception or natural language, respectively. See Table 1 for a comparison.

Talk The Walk is the first task to bring all three aspects together: *perception* for the tourist observing the world, *action* for the tourist to navigate through the environment, and *interactive dialogue* for the tourist and guide to work towards their common goal. To collect grounded dialogues, we constructed a virtual 2D grid environment by manually capturing 360-views of several neighbor-

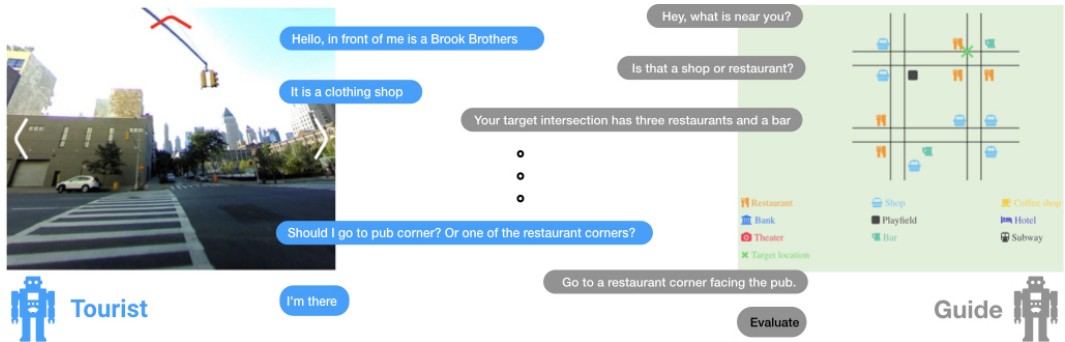

Figure 1: Example of the Talk The Walk task: two agents, a "tourist" and a "guide", interact with each other via natural language in order to have the tourist navigate towards the correct location. The guide has access to a map and knows the target location but not the tourist location, while the tourist does not have a map and is tasked with navigating a 360-degree street view environment.

hoods in New York City (NYC)[1]. As the main focus of our task is on interactive dialogue, we limit the difficulty of the control problem by having the tourist navigating a 2D grid via discrete actions (turning left, turning right and moving forward). Our street view environment was integrated into ParlAI (Miller et al., 2017) and used to collect a large-scale dataset on Mechanical Turk involving human perception, action and communication.

We argue that for artificial agents to solve this challenging problem, some fundamental architecture designs are missing, and our hope is that this task motivates their innovation. To that end, we focus on the task of localization and develop the novel Masked Attention for Spatial Convolutions (MASC) mechanism. To model the interaction between language and action, this architecture repeatedly conditions the spatial dimensions of a convolution on the communicated message sequence.

This work makes the following contributions: 1) We present the first large scale dialogue dataset grounded in action and perception; 2) We introduce the MASC architecture for localization and show it yields improvements for both emergent and natural language; 4) Using localization models, we establish initial baselines on the full task; 5) We show that our best model exceeds human performance under the assumption of "perfect perception" and with a learned emergent communication protocol, and sets a non-trivial baseline with natural language.

## 2 TALK THE WALK

We create a perceptual environment by manually capturing several neighborhoods of New York City (NYC) with a 360 camera[2]. Most parts of the city are grid-like and uniform, which makes it well-suited for obtaining a 2D grid. For Talk The Walk, we capture parts of Hell's Kitchen, East Village, the Financial District, Williamsburg and the Upper East Side—see Figure 5 in Appendix 13 for their respective locations within NYC. For each neighborhood, we choose an approximately 5x5 grid and capture a 360 view on all four corners of each intersection, leading to a grid-size of roughly 10x10 per neighborhood.

The tourist's location is given as a tuple $(x, y, o)$, where $x, y$ are the coordinates and $o$ signifies the orientation (north, east, south or west). The tourist can take three actions: turn left, turn right and go forward. For moving forward, we add $(0, 1)$, $(1, 0)$, $(0, -1)$, $(-1, 0)$ to the $x, y$ coordinates for the respective orientations. Upon a turning action, the orientation is updated by $o = (o + d) \mod 4$ where $d = -1$ for left and $d = 1$ for right. If the tourist moves outside the grid, we issue a warning that they cannot go in that direction and do not update the location. Moreover, tourists are shown different types of transitions: a short transition for actions that bring the tourist to a different corner of the *same* intersection; and a longer transition for actions that bring them to a new intersection.

The guide observes a map that corresponds to the tourist's environment. We exploit the fact that urban areas like NYC are full of local businesses, and overlay the map with these landmarks as localization points for our task. Specifically, we manually annotate each corner of the intersection with a set of landmarks $\Lambda^{x,y} = \{l_0, \ldots, l_K\}$, each coming from one of the following categories:

---

[1] We avoided using existing street view resources due to licensing issues.

[2] A 360fly 4K camera.

| Project | Perception | Action | Language | Dial. | Size | Acts |
|---|---|---|---|---|---|---|
| Visual Dialog (Das et al., 2016) | Real | ✗ | Human | ✓ | 120k dialogues | 20 |
| GuessWhat (de Vries et al., 2016) | Real | ✗ | Human | ✓ | 131k dialogues | 10 |
| VNL (Anderson et al., 2017) | Real | ✓ | Human | ✗ | 23k instructions | - |
| Embodied QA (Das et al., 2017a) | Simulated | ✓ | Scripted | ✗ | 5k questions | - |
| **TalkTheWalk** | Real | ✓ | Human | ✓ | 10k dialogues | 62 |

Table 1: Talk The Walk grounds human generated dialogue in (real-life) perception and action.

- Bar
- Playfield
- Bank
- Hotel
- Shop
- Subway
- Coffee Shop
- Restaurant
- Theater

The right-side of Figure 1 illustrates how the map is presented. Note that within-intersection transitions have a smaller grid distance than transitions to new intersections. To ensure that the localization task is not too easy, we do not include street names in the overhead map and keep the landmark categories coarse. That is, the dialogue is driven by uncertainty in the tourist's current location and the properties of the target location: if the exact location and orientation of the tourist were known, it would suffice to communicate a sequence of actions.

## 2.1 TASK

For the Talk The Walk task, we randomly choose one of the five neighborhoods, and subsample a 4x4 grid (one block with four complete intersections) from the entire grid. We specify the boundaries of the grid by the top-left and bottom-right corners $(x_{min}, y_{min}, x_{max}, y_{max})$. Next, we construct the overhead map of the environment, i.e. $\{\Lambda^{x',y'}\}$ with $x_{min} \leq x' \leq x_{max}$ and $y_{min} \leq y' \leq y_{max}$. We subsequently sample a start location and orientation $(x, y, o)$ and a target location $(x, y)_{tgt}$ at random[3].

The shared goal of the two agents is to navigate the tourist to the target location $(x, y)_{tgt}$, which is only known to the guide. The tourist perceives a "street view" planar projection $S_{x,y,o}$ of the 360 image at location $(x, y)$ and can simultaneously chat with the guide and navigate through the environment. The guide's role consists of reading the tourist description of the environment, building a "mental map" of their current position and providing instructions for navigating towards the target location. Whenever the guide believes that the tourist has reached the target location, they instruct the system to evaluate the tourist's location. The task ends when the evaluation is successful—i.e., when $(x, y) = (x, y)_{tgt}$—or otherwise continues until a total of *three* failed attempts. The additional attempts are meant to ease the task for humans, as we found that they otherwise often fail at the task but still end up close to the target location, e.g., at the wrong corner of the correct intersection.

## 2.2 DATA COLLECTION

We crowd-sourced the collection of the dataset on Amazon Mechanical Turk (MTurk). We use the MTurk interface of ParlAI (Miller et al., 2017) to render 360 images via WebGL and dynamically display neighborhood maps with an HTML5 canvas. Detailed task instructions, which were also given to our workers before they started their task, are shown in Appendix 14. We paired Turkers at random and let them alternate between the tourist and guide role across different HITs.

## 2.3 DATASET STATISTICS

The Talk The Walk dataset consists of over 10k successful dialogues—see Table 11 in the appendix for the dataset statistics split by neighborhood. Turkers successfully completed 76.74% of all finished tasks (we use this statistic as the human success rate). More than six hundred participants successfully completed at least one Talk The Walk HIT. Although the Visual Dialog (Das et al., 2016) and GuessWhat (de Vries et al., 2016) datasets are larger, the collected Talk The Walk dialogs are significantly longer. On average, Turkers needed more than 62 acts (i.e utterances and actions) before they successfully completed the task, whereas Visual Dialog requires 20 acts. The majority of acts comprise the tourist's actions, with on average more than 44 actions per dialogue. The guide produces roughly 9 utterances per dialogue, slightly more than the tourist's 8 utterances. Turkers use diverse discourse, with a vocabulary size of more than 10K (calculated over all successful dia-

---

[3]Note that we do not include the orientation in the target, as we found in early experiments that this led to an unnatural task for humans. Similarly, we explored bigger grid sizes but found these to be too difficult for most annotators.

logues). An example from the dataset is shown in Appendix 13. The dataset is available at [URL ANONYMIZED].

## 3   EXPERIMENTS

We investigate the difficulty of the proposed task by establishing initial baselines. The final Talk The Walk task is challenging and encompasses several important sub-tasks, ranging from landmark recognition to tourist localization and natural language instruction-giving. Arguably the most important sub-task is localization: without such capabilities the guide can not tell whether the tourist reached the target location. In this work, we establish a minimal baseline for Talk The Walk by utilizing agents trained for localization. Specifically, we let trained tourist models undertake random walks, using the following protocol: at each step, the tourist communicates its observations and actions to the guide, who predicts the tourist's location. If the guide predicts that the tourist is at target, we evaluate its location. If successful, the task ends, otherwise we continue until there have been three wrong evaluations. The protocol is given as pseudo-code in Appendix 11.

### 3.1   TOURIST LOCALIZATION

The designed navigation protocol relies on a trained localization model that predicts the tourist's location from a communicated message. Before we formalize this localization sub-task in Section 3.1.1, we further introduce two simplifying assumptions—perfect perception and orientation-agnosticism—so as to overcome some of the difficulties we encountered in preliminary experiments.

**Perfect Perception**   Early experiments revealed that perceptual grounding of landmarks is difficult: we set up a landmark classification problem, on which models with extracted CNN (He et al., 2016) or text recognition features (Gupta et al., 2016) barely outperform a random baseline—see Appendix 12 for full details. This finding implies that localization models from image input are limited by their ability to recognize landmarks, and, as a result, would not generalize to unseen environments. To ensure that perception is not the limiting factor when investigating the landmark-grounding and action-grounding capabilities of localization models, we assume "perfect perception": in lieu of the 360 image view, the tourist is given the landmarks at its current location. More formally, each state observation $S^{x,y,o}$ now equals the set of landmarks at the $(x,y)$-location, i.e. $S^{x,y,o} = \Lambda^{x,y}$. If the $(x,y)$-location does not have any visible landmarks, we return a single "empty corner" symbol. We stress that our findings—including a novel architecture for grounding actions into an overhead map, see Section 4.2.1—should carry over to settings without the perfect perception assumption.

**Orientation-agnosticism**   We opt to ignore the tourist's orientation, which simplifies the set of actions to [Left, Right, Up, Down], corresponding to adding [(-1, 0), (1, 0), (0, 1), (0, -1)] to the current $(x,y)$ coordinates, respectively. Note that actions are now coupled to an orientation on the map—e.g. up is equal to going north—and this implicitly assumes that the tourist has access to a compass. This also affects perception, since the tourist now has access to views from all orientations: in conjunction with "perfect perception", implying that only landmarks at the current corner are given, whereas landmarks from different corners (e.g. across the street) are not visible.

Even with these simplifications, the localization-based baseline comes with its own set of challenges. As we show in Section 5.1, the task requires communication about a short (random) path—i.e., not only a sequence of observations but also *actions*—in order to achieve high localization accuracy. This means that the guide needs to decode observations from multiple time steps, as well as understand their 2D spatial arrangement as communicated via the sequence of actions. Thus, in order to get to a good understanding of the task, we thoroughly examine whether the agents can learn a communication protocol that simultaneously grounds observations and actions into the guide's map. In doing so, we thoroughly study the role of the communication channel in the localization task, by investigating increasingly constrained forms of communication: from differentiable continuous vectors to emergent discrete symbols to the full complexity of natural language.

### 3.1.1   FORMALIZATION

The full navigation baseline hinges on a localization model from random trajectories. While we can sample random actions in the emergent communication setup, this is not possible for the natural language setup because the messages are coupled to the trajectories of the human annotators. This leads to slightly different problem setups, as described below.

**Emergent language** A tourist, starting from a random location, takes $T \geq 0$ *random* actions $A = \{\alpha_0, \ldots, \alpha_{T-1}\}$ to reach target location $(x_{tgt}, y_{tgt})$. Every location in the environment has a corresponding set of landmarks $\Lambda^{x,y} = \{l_0, \ldots, l_K\}$ for each of the $(x, y)$ coordinates. As the tourist navigates, the agent perceives $T + 1$ state-observations $Z = \{\zeta_0, \ldots, \zeta_T\}$ where each observation $\zeta_t$ consists of a set of $K$ landmark symbols $\{l_0^t, \ldots, l_K^t\}$. Given the observations $Z$ and actions $A$, the tourist generates a message $M$ which is communicated to the other agent. The objective of the guide is to predict the location $(x_{tgt}, y_{tgt})$ from the tourist's message $M$.

**Natural language** In contrast to our emergent communication experiments, we do not take random actions but instead extract actions, observations, and messages from the dataset. Specifically, we consider each tourist utterance (i.e. at any point in the dialogue), obtain the current tourist location as target location $(x, y)_{tgt}$, the utterance itself as message $M$, and the sequence of observations and actions that took place between the current and previous tourist utterance as $Z$ and $A$, respectively. Similar to the emergent language setting, the guide's objective is to predict the target location $(x, y)_{tgt}$ models from the tourist message $M$. We conduct experiments with $M$ taken from the dataset and with $M$ generated from the extracted observations $Z$ and actions $A$.

## 4 MODEL

This section outlines the tourist and guide architectures. We first describe how the tourist produces messages for the various communication channels across which the messages are sent. We subsequently describe how these messages are processed by the guide, and introduce the novel Masked Attention for Spatial Convolutions (MASC) mechanism that allows for grounding into the 2D overhead map in order to predict the tourist's location.

### 4.1 THE TOURIST

For each of the communication channels, we outline the procedure for generating a message $M$. Given a set of state observations $\{\zeta_0, \ldots, \zeta_T\}$, we represent each observation by summing the $L$-dimensional embeddings of the observed landmarks, i.e. for $\{\mathbf{o}_0, \ldots, \mathbf{o}_T\}$, $\mathbf{o}_t = \sum_{l \in \zeta_t} E^\Lambda(l)$, where $E^\Lambda$ is the landmark embedding lookup table. In addition, we embed action $\alpha_t$ into a $L$-dimensional embedding $\mathbf{a}_t$ via a look-up table $E^A$. We experiment with three types of communication channel.

**Continuous vectors** The tourist has access to observations of several time steps, whose order is important for accurate localization. Because summing embeddings is order-invariant, we introduce a sum over *positionally-gated* embeddings, which, conditioned on time step $t$, pushes embedding information into the appropriate dimensions. More specifically, we generate an observation message $\mathbf{m}^{obs} = \sum_{t=0}^{T} \text{sigmoid}(\mathbf{g}_t) \odot \mathbf{o}_t$, where $\mathbf{g}_t$ is a learned gating vector for time step $t$. In a similar fashion, we produce action message $\mathbf{m}^{act}$ and send the concatenated vectors $\mathbf{m} = [\mathbf{m}^{obs}; \mathbf{m}^{act}]$ as message to the guide. We can interpret continuous vector communication as a single, monolithic model because its architecture is end-to-end differentiable, enabling gradient-based optimization for training.

**Discrete symbols** Like the continuous vector communication model, with discrete communication the tourist also uses separate channels for observations and actions, as well as a sum over positionally gated embeddings to generate observation embedding $\mathbf{h}^{obs}$. We pass this embedding through a sigmoid and generate a message $\mathbf{m}^{obs}$ by sampling from the resulting Bernoulli distributions:

$$\mathbf{h}^{obs} = \sum_{t=0}^{T} \text{sigmoid}(\mathbf{g}_t) \odot \mathbf{o}_t; \qquad \mathbf{m}_i^{obs} \sim \texttt{Bernoulli}(\text{sigmoid}(h_i^{obs}))$$

The action message $\mathbf{m}^{act}$ is produced in the same way, and we obtain the final tourist message $\mathbf{m} = [\mathbf{m}^{obs}; \mathbf{m}^{act}]$ through concatenating the messages.

The communication channel's sampling operation yields the model non-differentiable, so we use policy gradients (Sutton & Barto, 1998; Williams, 1992) to train the parameters $\theta$ of the tourist model. That is, we estimate the gradient by

$$\nabla_\theta E_{\mathbf{m} \sim p(\mathbf{h})}[r(\mathbf{m})] = E_{\mathbf{m}}[\nabla_\theta \log p(\mathbf{m})(r(\mathbf{m}) - b)],$$

where the reward function $r(\mathbf{m}) = -\log p(x, y)_{tgt} | \mathbf{m}, \Lambda)$ is the negative guide's loss (see Section 4.2) and $b$ a state-value baseline to reduce variance. We use a linear transformation over the con-

catenated embeddings as baseline prediction, i.e. $b = W^{base}[\mathbf{h}^{obs}; \mathbf{h}^{act}] + \mathbf{b}^{base}$, and train it with a mean squared error loss[4].

**Natural Language**   Because observations and actions are of variable-length, we use an LSTM encoder over the sequence of observations embeddings $[\mathbf{o}_t]_{t=0}^{T+1}$, and extract its last hidden state $\mathbf{h}^{obs}$. We use a separate LSTM encoder for action embeddings $[\mathbf{a}_t]_{t=0}^{T}$, and concatenate both $\mathbf{h}^{obs}$ and $\mathbf{h}^{act}$ to the input of the LSTM decoder at each time step:

$$\mathbf{i}_k = [E^{dec}(w_{k-1}); \mathbf{h}^{obs}; \mathbf{h}^{act}] \qquad \mathbf{h}_k^{dec} = f_{LSTM}(\mathbf{i}_t, \mathbf{h}_{k-1}^{dec})$$
$$p(w_k | w_{<k}, A, Z) = \text{softmax}(W^{out}\mathbf{h}_k^{dec} + \mathbf{b}^{out})_k, \qquad (1)$$

where $E^{dec}$ a look-up table, taking input tokens $w_k$. We train with teacher-forcing, i.e. we optimize the cross-entropy loss: $-\sum_K \log p(w_k | w_{<k}, A, Z)$. At test time, we explore the following decoding strategies: greedy, sampling and a beam-search. We also fine-tune a trained tourist model (starting from a pre-trained model) with policy gradients in order to minimize the guide's prediction loss.

## 4.2   THE GUIDE

Given a tourist message $M$ describing their observations and actions, the objective of the guide is to predict the tourist's location on the map. First, we outline the procedure for extracting observation embedding $\mathbf{e}$ and action embeddings $\mathbf{a}_t$ from the message $M$ for each of the types of communication. Next, we discuss the MASC mechanism that takes the observations and actions in order to ground them on the guide's map in order to predict the tourist's location.

**Continuous**   For the continuous communication model, we assign the observation message to the observation embedding, i.e. $\mathbf{e} = \mathbf{m}^{obs}$. To extract the action embedding for time step $t$, we apply a linear layer to the action message, i.e. $\mathbf{a}_t = W_t^{act}\mathbf{m}^{act} + \mathbf{b}_t^{act}$.

**Discrete**   For discrete communication, we obtain observation $\mathbf{e}$ by applying a linear layer to the observation message, i.e. $\mathbf{e} = W^{obs}\mathbf{m}^{obs} + \mathbf{b}^{obs}$. Similar to the continuous communication model, we use a linear layer over action message $\mathbf{m}^{act}$ to obtain action embedding $\mathbf{a}_t$ for time step $t$.

**Natural Language**   The message $M$ contains information about observations and actions, so we use a recurrent neural network with attention mechanism to extract the relevant observation and action embeddings. Specifically, we encode the message $M$, consisting of $K$ tokens $w_k$ taken from vocabulary $V$, with a bidirectional LSTM:

$$\overrightarrow{\mathbf{h}_k} = f_{LSTM}(\overrightarrow{\mathbf{h}_{k-1}}, E^W(w_k)); \qquad \overleftarrow{\mathbf{h}_k} = f_{LSTM}(\overleftarrow{\mathbf{h}_{k+1}}, E^W(w_k)); \qquad \mathbf{h}_k = [\overrightarrow{\mathbf{h}_k}; \overleftarrow{\mathbf{h}_k}] \quad (2)$$

where $E^W$ is the word embedding look-up table. We obtain observation embedding $\mathbf{e}_t$ through an attention mechanism over the hidden states $\mathbf{h}$:

$$s_k = \mathbf{h}_k \cdot \mathbf{c}_t; \quad \mathbf{e}_t = \sum_k \text{softmax}(\mathbf{s})_k \mathbf{h}_k, \qquad (3)$$

where $\mathbf{c}_0$ is a learned control embedding who is updated through a linear transformation of the previous control and observation embedding: $\mathbf{c}_{t+1} = W^{ctrl}[\mathbf{c}_t; \mathbf{e}_t] + \mathbf{b}^{ctrl}$. We use the same mechanism to extract the action embedding $\mathbf{a}_t$ from the hidden states. For the observation embedding, we obtain the final representation by summing positionally gated embeddings, i.e., $\mathbf{e} = \sum_{t=0}^{T} = \text{sigmoid}(\mathbf{g}_t) \odot \mathbf{e}_t$.

### 4.2.1   MASKED ATTENTION FOR SPATIAL CONVOLUTIONS (MASC)

We represent the guide's map as $U \in \mathbb{R}^{G_1 \times G_2 \times L}$, where in this case $G_1 = G_2 = 4$, where each $L$-dimensional $(x, y)$ location embedding $\mathbf{u}^{x,y}$ is computed as the sum of the guide's landmark embeddings for that location.

**Motivation**   While the guide's map representation contains only local landmark information, the tourist communicates a trajectory of the map (i.e. actions and observations from multiple locations), implying that directly comparing the tourist's message with the individual landmark embeddings is probably suboptimal. Instead, we want to aggregate landmark information from surrounding locations by imputing trajectories over the map to predict locations. We propose a mechanism for

---

[4]This is different from A2C which uses a state-value baseline that is trained by the Bellman residual

| | MASC | Train | | | Valid | | | Test | | |
|---|---|---|---|---|---|---|---|---|---|---|
| | | Cont. | Disc. | Upper | Cont. | Disc. | Upper | Cont. | Disc. | Upper |
| Random | | 6.25 | 6.25 | 6.25 | 6.25 | 6.25 | 6.25 | 6.25 | 6.25 | 6.25 |
| T=0 | ✗ | 29.59 | 28.89 | 30.23 | 30.00 | 30.63 | 32.50 | 32.29 | 33.12 | 35.00 |
| T=1 | ✗ | 39.83 | 35.40 | 43.44 | 35.23 | 36.56 | 45.39 | 35.16 | 39.53 | 51.72 |
| | ✓ | 55.64 | 51.66 | 62.78 | 53.12 | 53.20 | 65.78 | 56.09 | 55.78 | 72.97 |
| T=2 | ✗ | 41.50 | 40.15 | 47.84 | 33.50 | 37.77 | 50.29 | 35.08 | 41.41 | 57.15 |
| | ✓ | 67.44 | 62.24 | 78.90 | 64.55 | 59.34 | 79.77 | 66.80 | 62.15 | 86.64 |
| T=3 | ✗ | 43.48 | 44.49 | 45.22 | 35.40 | 39.64 | 48.77 | 33.11 | 43.51 | 55.84 |
| | ✓ | 71.32 | 71.80 | 87.92 | 67.48 | 65.63 | 87.45 | 69.85 | 69.51 | 92.41 |

Table 2: Accuracy results for tourist localization with emergent language, showing continuous (Cont.) and discrete (Disc.) communication, along with the prediction upper bound. $T$ denotes the length of the path and a ✓ in the "MASC" column indicates that the model is conditioned on the communicated actions.

translating landmark embeddings according to state transitions (left, right, up, down), which can be expressed as a 2D convolution over the map embeddings. For simplicity, let us assume that the map embedding $U$ is 1-dimensional, then a left action can be realized through application of the following 3x3 kernel: $\begin{smallmatrix} 0 & 0 & 0 \\ 1 & 0 & 0 \\ 0 & 0 & 0 \end{smallmatrix}$, which effectively shifts all values of $U$ one position to the left. We propose to learn such state-transitions from the tourist message through a differentiable attention-mask over the spatial dimensions of a 3x3 convolution.

**MASC**  We linearly project each predicted action embedding $\mathbf{a}_t$ to a 9-dimensional vector $\mathbf{z}_t$, normalize it by a softmax and subsequently reshape the vector into a 3x3 mask $\Phi_t$:

$$\mathbf{z}_t = W^{act}\mathbf{a}_t + b^{act}, \qquad \phi_t = \text{softmax}(\mathbf{z}_t), \qquad \Phi_t = \begin{bmatrix} \phi_t^0 & \phi_t^1 & \phi_t^2 \\ \phi_t^3 & \phi_t^4 & \phi_t^5 \\ \phi_t^6 & \phi_t^7 & \phi_t^8 \end{bmatrix}. \qquad (4)$$

We learn a 3x3 convolutional kernel $W \in \mathbb{R}^{3 \times 3 \times N \times N}$, with $N$ features, and apply the mask $\Phi_t$ to the spatial dimensions of the convolution by first broadcasting its values along the feature dimensions, i.e. $\hat{\Phi}^{x,y,i,j} = \Phi^{x,y}$, and subsequently taking the Hadamard product: $W_t = \hat{\Phi}_t \odot W$. For each action step $t$, we then apply a 2D convolution with masked weight $W_t$ to obtain a new map embedding $U_{t+1} = U_t * W_t$, where we zero-pad the input to maintain identical spatial dimensions.

**Prediction model**  We repeat the MASC operation $T$ times (i.e. once for each action), and then aggregate the map embeddings by a sum over positionally-gated embeddings: $\mathbf{u}^{x,y} = \sum_{t=0}^{T} \text{sigmoid}(\mathbf{g}_t) \odot \mathbf{u}_t^{x,y}$. We score locations by taking the dot-product of the observation embedding $\mathbf{e}$, which contains information about the sequence of observed landmarks by the tourist, and the map. We compute a distribution over the locations of the map $p(x, y|M, \Lambda)$ by taking a softmax over the computed scores:

$$s^{x,y} = \mathbf{e} \cdot \mathbf{u}^{x,y}, \qquad p(x, y|M, \Lambda) = \frac{\exp(s^{x,y})}{\sum_{x',y'} \exp(s^{x',y'})}. \qquad (5)$$

**Predicting T**  While emergent communication models use a fixed length trasjectory $T$, natural language messages may differ in the number of communicated observations and actions. Hence, we predict $T$ from the communicated message. Specifically, we use a softmax regression layer over the last hidden state $\mathbf{h}_K$ of the RNN, and subsequently sample $T$ from the resulting multinomial distribution:

$$\mathbf{z} = \text{softmax}(W^{tm}\mathbf{h}_K + \mathbf{b}^{tm}); \qquad \hat{T} \sim \text{Multinomial}(\mathbf{z}). \qquad (6)$$

We jointly train the $T$-prediction model via REINFORCE, with the guide's loss as reward function and a mean-reward baseline.

### 4.3 Comparisons

To better analyze the performance of the models incorporating MASC, we compare against a no-MASC baseline in our experiments, as well as a prediction upper bound.

**No MASC**    We compare the proposed MASC model with a model that does not include this mechanism. Whereas MASC predicts a convolution mask from the tourist message, the "No MASC" model uses $W$, the ordinary convolutional kernel to convolve the map embedding $U_t$ to obtain $U_{t+1}$. We also share the weights of this convolution at each time step.

**Prediction upper-bound**    Because we have access to the class-conditional likelihood $p(Z, A|x, y)$, we are able to compute the Bayes error rate (or irreducible error). No model (no matter how expressive) with any amount of data can ever obtain better localization accuracy as there are multiple locations consistent with the observations and actions.

## 5   RESULTS AND DISCUSSION

In this section, we describe the findings of various experiments. First, we analyze how much information needs to be communicated for accurate localization in the Talk The Walk environment, and find that a short random path (including actions) is necessary. Next, for emergent language, we show that the MASC architecture can achieve very high localization accuracy, significantly outperforming the baseline that does not include this mechanism. We then turn our attention to the natural language experiments, and find that localization from human utterances is much harder, reaching an accuracy level that is below communicating a single landmark observation. We show that generated utterances from a conditional language model leads to significantly better localization performance, by successfully grounding the utterance on a single landmark observation (but not yet on multiple observations and actions). Finally, we show performance of the localization baseline on the full task, which can be used for future comparisons to this work.

### 5.1   ANALYSIS OF LOCALIZATION TASK

**Task is not too easy**    The upper-bound on localization performance in Table 2 suggest that communicating a single landmark observation is not sufficient for accurate localization of the tourist ($\sim 35\%$ accuracy). This is an important result for the full navigation task because the need for two-way communication disappears if localization is too easy; if the guide knows the exact location of the tourist it suffices to communicate a list of instructions, which is then executed by the tourist. The uncertainty in the tourist's location is what drives the dialogue between the two agents.

**Importance of actions**    We observe that the upperbound for only communicating observations plateaus around $57\%$ (even for $T = 3$ actions), whereas it exceeds $90\%$ when we also take actions into account. This implies that, at least for random walks, it is essential to communicate a trajectory, including observations and *actions*, in order to achieve high localization accuracy.

### 5.2   EMERGENT LANGUAGE LOCALIZATION

We first report the results for tourist localization with **emergent language** in Table 2.

**MASC improves performance**    The MASC architecture significantly improves performance compared to models that do not include this mechanism. For instance, for $T = 1$ action, MASC already achieves 56.09 % on the test set and this further increases to 69.85% for $T = 3$. On the other hand, no-MASC models hit a plateau at 43%. In Appendix 10, we analyze learned MASC values, and show that communicated actions are often mapped to corresponding state-transitions.

**Continuous vs discrete**    We observe similar performance for continuous and discrete emergent communication models, implying that a discrete communication channel is not a limiting factor for localization performance.

### 5.3   NATURAL LANGUAGE LOCALIZATION

We report the results of tourist localization with **natural language** in Table 3. We compare accuracy of the guide model (with MASC) trained on utterances from (i) humans, (ii) a supervised model with various decoding strategies, and (iii) a policy gradient model optimized with respect to the loss of a frozen, pre-trained guide model on *human* utterances.

**Human utterances**    Compared to emergent language, localization from human utterances is much harder, achieving only 16.17% on the test set. Here, we report localization from a single utterance, but in Appendix 9.2 we show that including up to five dialogue utterances only improves performance to 20.33%. We also show that MASC outperform no-MASC models for natural language communication.

| Model | Decoding | Train | Valid | Test | | Train | Valid | Test | #steps |
|-------|----------|-------|-------|------|------|-------|-------|------|--------|
| Random | | 6.25 | 6.25 | 6.25 | Random | 18.75 | 18.75 | 18.75 | - |
| Human utterances | | 23.46 | 15.56 | 16.17 | Human | 76.74 | 76.74 | 76.74 | 15.05 |
| Supervised | sampling | 17.19 | 12.23 | 12.43 | Best Cont. | 89.44 | 86.35 | 88.33 | 34.47 |
| | greedy | 34.14 | 29.90 | 29.05 | Best Disc. | 86.23 | 82.81 | 87.08 | 34.83 |
| | beam (size: 4) | 26.21 | 22.53 | 25.02 | Best NL | 39.65 | 39.68 | 50.00 | 39.14 |
| Policy Grad. | sampling | 29.67 | 26.93 | 27.05 | | | | | |
| | greedy | 29.23 | 27.62 | 27.30 | | | | | |

Table 3: **Localization** accuracy of tourist communicating in natural language.

Table 4: **Full task** evaluation of localization models using protocol of Appendix 11.

| Method | Decoding | Utterance |
|--------|----------|-----------|
| Observations | | (Bar) |
| Actions | | - |
| Human | | a field of some type |
| Supervised | greedy | at a bar |
| | sampling | sec just hard to tell which is a restaurant ? |
| | beam search | im at a bar |
| Policy Grad. | greedy | bar from bar from bar and rigth rigth bulding buling |
| | sampling | which bar from bar from bar and bar rigth bulding buling.. |

Table 5: Samples from the tourist models communicating in natural language. Contrary to the human generated utterance, the supervised model with greedy and beam search decoding produces an utterance containing the current state observation (bar). Also the reinforcement learning model mentions the current observation but has lost linguistic structure. The fact that these localization models are better grounded in observations than human utterances explains why they obtain higher localization accuracy.

**Generated utterances** We also investigate generated tourist utterances from conditional language models. Interestingly, we observe that the supervised model (with greedy and beam-search decoding) as well as the policy gradient model leads to an improvement of more than 10 accuracy points over the human utterances. However, their level of accuracy is slightly below the baseline of communicating a single observation, indicating that these models only learn to ground utterances in a single landmark observation.

**Better grounding of generated utterances** We analyze natural language samples in Table 5, and confirm that, unlike human utterances, the generated utterances are talking about the observed landmarks. This observation explains why the generated utterances obtain higher localization accuracy. The current language models are most successful when conditioned on a single landmark observation; We show in Appendix 9.1.1 that performance quickly deteriorates when the model is conditioned on more observations, suggesting that it can not produce natural language utterances about multiple time steps.

### 5.4 LOCALIZATION-BASED BASELINE

Table 4 shows results for the best localization models on the **full task**, evaluated via the random walk protocol defined in Algorithm 1.

**Comparison with human annotators** Interestingly, our best localization model (continuous communication, with MASC, and $T = 3$) achieves 88.33% on the test set and thus exceed human performance of 76.74% on the full task. While emergent models appear to be stronger localizers, humans might cope with their localization uncertainty through other mechanisms (e.g. better guidance, bias towards taking particular paths, etc). The simplifying assumption of perfect perception also helps.

**Number of actions** Unsurprisingly, humans take fewer steps (roughly 15) than our best random walk model (roughly 34). Our human annotators likely used some form of guidance to navigate faster to the target.

## 6 CONCLUSION

We introduced the Talk The Walk task and dataset, which consists of crowd-sourced dialogues in which two human annotators collaborate to navigate to target locations in the virtual streets of NYC. For the important localization sub-task, we proposed MASC—a novel grounding mechanism to learn state-transition from the tourist's message—and showed that it improves localization performance for emergent and natural language. We use the localization model to provide baseline numbers on the Talk The Walk task, in order to facilitate future research.

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

## 7 RELATED WORK

The Talk the Walk task and dataset facilitate future research on various important subfields of artificial intelligence, including grounded language learning, goal-oriented dialogue research and situated navigation. Here, we describe related previous work in these areas.

**Related tasks** There has been a long line of work involving related tasks. Early work on task-oriented dialogue dates back to the early 90s with the introduction of the Map Task (Anderson et al., 1991) and Maze Game (Garrod & Anderson, 1987) corpora. Recent efforts have led to larger-scale goal-oriented dialogue datasets, for instance to aid research on visually-grounded dialogue (Das et al., 2016; de Vries et al., 2016), knowledge-base-grounded discourse (He et al., 2017) or negotiation tasks (Lewis et al., 2017). At the same time, there has been a big push to develop environments for embodied AI, many of which involve agents following natural language instructions with respect to an environment(Artzi & Zettlemoyer, 2013; Yu et al., 2017; Hermann et al., 2017; Mei et al., 2016; Chaplot et al., 2018b;a), following-up on early work in this area (MacMahon et al., 2006; Chen & Mooney, 2011). An early example of navigation using neural networks is (Hadsell et al., 2007), who propose an online learning approach for robot navigation. Recently, there has been increased interest in using end-to-end trainable neural networks for learning to navigate indoor scenes(Gupta et al., 2017b;a) or large cities (Brahmbhatt & Hays, 2017; Mirowski et al., 2018), but, unlike our work, without multi-agent communication. Also the task of localization (without multi-agent communication) has recently been studied (Chaplot et al., 2018a; Vo et al., 2017).

**Grounded language learning** Grounded language learning is motivated by the observation that humans learn language embodied (grounded) in sensorimotor experience of the physical world (Barsalou, 2008; Smith & Gasser, 2005). On the one hand, work in multi-modal semantics has shown that grounding can lead to practical improvements on various natural language understanding tasks (see Baroni, 2016; Kiela, 2017, and references therein). In robotics, researchers dissatisfied with purely symbolic accounts of meaning attempted to build robotic systems with the aim of grounding meaning in physical experience of the world (Roy, 2005; Steels & Hild, 2012). Recently, grounding has also been applied to the learning of sentence representations (Kiela et al., 2017), image captioning (Lin et al., 2014; Xu et al., 2015), visual question answering (Antol et al., 2015; de Vries et al., 2017), visual reasoning (Johnson et al., 2017; Perez et al., 2018), and grounded machine translation (Riezler et al., 2014; Elliott et al., 2016). Grounding also plays a crucial role in the emergent research of multi-agent communication, where, agents communicate (in natural language or otherwise) in order to solve a task, with respect to their shared environment (Lazaridou et al., 2016; Das et al., 2017b; Mordatch & Abbeel, 2017; Evtimova et al., 2017; Lewis et al., 2017; Strub et al., 2017; Kottur et al., 2017).

## 8 IMPLEMENTATION DETAILS

For the emergent communication models, we use an embedding size $L = 500$. The natural language experiments use 128-dimensional word embeddings and a bidirectional RNN with 256 units. In all experiments, we train the guide with a cross entropy loss using the ADAM optimizer with default hyper-parameters (Kingma & Ba, 2014). We perform early stopping on the validation accuracy, and report the corresponding train, valid and test accuracy. We optimize the localization models with continuous, discrete and natural language communication channels for 200, 200, and 25 epochs, respectively. To facilitate further research on Talk The Walk, we make our code base for reproducing experiments publicly available at [URL ANONYMIZED].

## 9 ADDITIONAL NATURAL LANGUAGE EXPERIMENTS

First, we investigate the sensitivity of tourist generation models to the trajectory length, finding that the model conditioned on a single observation (i.e. $T = 0$) achieves best performance. In the next subsection, we further analyze localization models from human utterances by investigating MASC and no-MASC models with increasing dialogue context.

### 9.1 TOURIST GENERATION MODELS

#### 9.1.1 PATH LENGTH

After training the supervised tourist model (conditioned on observations and action from human expert trajectories), there are two ways to train an accompanying guide model. We can optimize

| Trajectories | T | Train | Valid | Test |
|---|---|---|---|---|
| Random | | 18.75 | 18.75 | 18.75 |
| Human | 0 | 38.21 | 40.93 | 40.00 |
| | 1 | 21.82 | 23.75 | 25.62 |
| | 2 | 19.77 | 24.68 | 23.12 |
| | 3 | 18.95 | 20.93 | 20.00 |
| Random | 0 | 39.65 | 39.68 | 50.00 |
| | 1 | 28.99 | 30.93 | 25.62 |
| | 2 | 27.04 | 19.06 | 19.38 |
| | 3 | 20.28 | 20.93 | 22.50 |

| Beam size | Train | Valid | Test |
|---|---|---|---|
| Random | 6.25 | 6.25 | 6.25 |
| 1 | 34.14 | 29.90 | 29.05 |
| 2 | 26.24 | 23.65 | 25.10 |
| 4 | 23.59 | 22.87 | 21.80 |
| 8 | 20.31 | 19.24 | 20.87 |

Table 6: Full task performance of localization models trained on human and random trajectories. There are small benefits for training on random trajectories, but the most important hyperparameter is to condition the tourist utterance on a single observation (i.e. trajectories of size $T = 0$.) at evaluation time.

Table 7: Localization performance using pretrained tourist (via imitation learning) with beam search decoding of varying beam size. Locations and observations extracted from human trajectories. Larger beam-sizes lead to worse localization performance.

| #utterances | MASC | Train | Valid | Test | $E[T]$ |
|---|---|---|---|---|---|
| Random | | 6.25 | 6.25 | 6.25 | - |
| 1 | ✗ | 23.95 | 13.91 | 13.89 | 0.99 |
| | ✓ | 23.46 | 15.56 | 16.17 | 1.00 |
| 3 | ✗ | 26.92 | 16.28 | 16.62 | 1.00 |
| | ✓ | 20.88 | 17.50 | 18.80 | 1.79 |
| 5 | ✗ | 25.75 | 16.11 | 16.88 | 1.98 |
| | ✓ | 30.45 | 18.41 | 20.33 | 1.99 |

Table 8: Localization given last $\{1, 3, 5\}$ dialogue utterances (including the guide). We observe that (1) performance increases when more utterances are included; and (2) MASC outperforms no-MASC in all cases; and (3) mean $\hat{T}$ increases when more dialogue context is included.

a location prediction model on either (i) extracted human trajectories (as in the localization setup from human utterances) or (ii) on all random paths of length $T$ (as in the full task evaluation). Here, we investigate the impact of (1) using either human or random trajectories for training the guide model, and (2) the effect of varying the path length $T$ during the full-task evaluation. For random trajectories, guide training uses the same path length $T$ as is used during evaluation. We use a pretrained tourist model with greedy decoding for generating the tourist utterances. Table 6 summarizes the results.

**Human vs random trajectories**  We only observe small improvements for training on random trajectories. Human trajectories are thus diverse enough to generalize to random trajectories.

**Effect of path length**  There is a strong negative correlation between task success and the conditioned trajectory length. We observe that the full task performance quickly deteriorates for both human and random trajectories. This suggests that the tourist generation model can not produce natural language utterances that describe multiple observations and actions. Although it is possible that the guide model can not process such utterances, this is not very likely because the MASC architectures handles such messages successfully for emergent communication.

### 9.1.2 EFFECT OF BEAM-SIZE

We report localization performance of tourist utterances generated by beam search decoding of *varying beam size* in Table 7. We find that performance decreases from 29.05% to 20.87% accuracy on the test set when we increase the beam-size from one to eight.

## 9.2 LOCALIZATION FROM HUMAN UTTERANCES

We conduct an ablation study for MASC on natural language with varying dialogue context. Specifically, we compare localization accuracy of MASC and no-MASC models trained on the last [1, 3, 5] utterances of the dialogue (including guide utterances). We report these results in Table 8. In all cases, MASC outperforms the no-MASC models by several accuracy points. We also observe that mean predicted $\hat{T}$ (over the test set) increases from 1 to 2 when more dialogue context is included.

## 10 VISUALIZING MASC PREDICTIONS

Figure 2 shows the MASC values for a learned model with emergent discrete communications and $T = 3$ actions. Specifically, we look at the predicted MASC values for different action sequences taken by the tourist. We observe that the first action is always mapped to the correct state-transition, but that the second and third MASC values do not always correspond to right state-transitions.

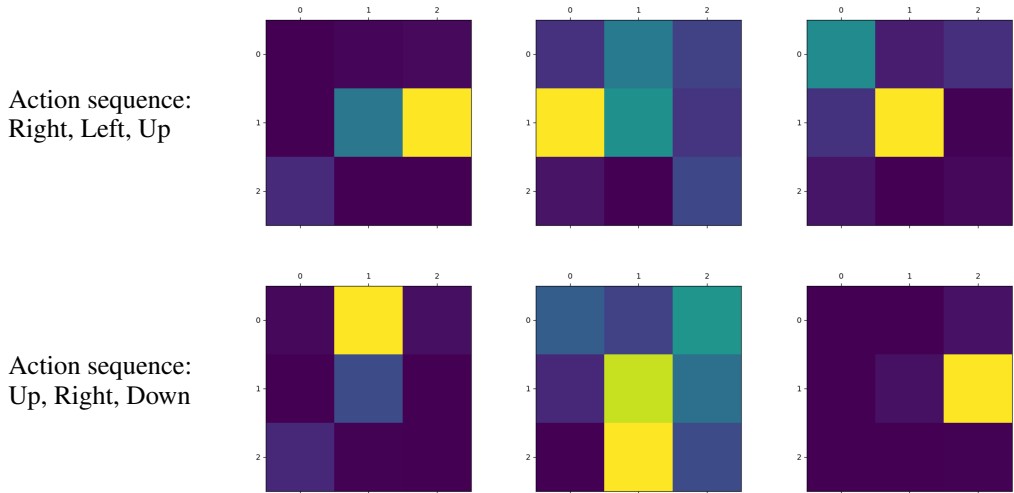

Figure 2: We show MASC values of two action sequences for tourist localization via *discrete* communication with $T = 3$ actions. In general, we observe that the first action always corresponds to the correct state-transition, whereas the second and third are sometimes mixed. For instance, in the top example, the first two actions are correctly predicted but the third action is not (as the MASC corresponds to a "no action"). In the bottom example, the second action appears as the third MASC.

## 11  EVALUATION ON FULL SETUP

We provide pseudo-code for evaluation of localization models on the full task in Algorithm 1, as well as results for all emergent communication models in Table 9.

| T | MASC | Comm. | Train | Valid | Test | #steps |
|---|------|-------|-------|-------|------|--------|
| Random | | | 18.75 | 18.75 | 18.75 | - |
| Human | | | 76.74 | 76.74 | 76.74 | 15.05 |
| 0 | ✗ | cont. | 46.17 | 46.56 | 52.91 | 39.87 |
| | | disc. | 46.65 | 47.70 | 52.50 | 38.56 |
| 1 | ✗ | cont. | 51.46 | 46.98 | 46.46 | 38.05 |
| | | disc | 52.11 | 51.25 | 55.00 | 41.13 |
| | ✓ | cont. | 76.57 | 74.06 | 77.70 | 34.59 |
| | | disc | 71.96 | 72.29 | 74.37 | 36.19 |
| 2 | ✗ | cont. | 53.51 | 45.93 | 46.66 | 40.26 |
| | | disc | 53.38 | 52.39 | 55.00 | 42.35 |
| | ✓ | cont. | 87.29 | 84.05 | 86.66 | 32.27 |
| | | disc | 86.23 | 82.81 | 87.08 | 34.83 |
| 3 | ✗ | cont. | 54.30 | 43.43 | 43.54 | 39.14 |
| | | disc | 57.88 | 55.20 | 57.50 | 43.67 |
| | ✓ | cont. | 89.44 | 86.35 | 88.33 | 34.47 |
| | | disc | 86.23 | 82.81 | 87.08 | 34.83 |

Table 9: Accuracy of localization models on full task, using evaluation protocol defined in Algorithm 1. We report the average over 3 runs.

---

**Algorithm 1** Performance evaluation of location prediction model on full Talk The Walk setup

---

**procedure** EVALUATE(tourist, guide, $T$, $x_{tgt}$, $y_{tgt}$, maxsteps)
    $x, y \leftarrow \mathrm{randint}(0, 3), \mathrm{randint}(0, 3)$                       $\triangleright$ initialize with random location
    $\mathrm{features}, \mathrm{actions} \leftarrow \mathrm{array}(), \mathrm{array}()$
    $\mathrm{features}[0] \leftarrow$ features at location (x, y)
    **for** $t = 0; t < T; t++$ **do**                       $\triangleright$ create $T$-sized feature buffer
        $\mathrm{action} \leftarrow$ uniform sample from action set
        $x, y \leftarrow$ update location given action
        $\mathrm{features}[t + 1] \leftarrow$ features at location (x, y)
        $\mathrm{actions}[t] \leftarrow \mathrm{action}$

    **for** $i = 0; i < \mathrm{maxsteps}; i++$ **do**
        $M \leftarrow \mathrm{tourist}(\mathrm{features}, \mathrm{actions})$
        $p(x, y|\cdot) \leftarrow \mathrm{guide}(M)$
        $x_{pred}, y_{pred} \leftarrow$ sample from $p(x, y|\cdot)$
        **if** $x_{pred}, y_{pred} == x_{tgt}, y_{tgt}$ **then**                $\triangleright$ target predicted
            **if** $\mathrm{locations}[0] == x_{tgt}, y_{tgt}$ **then**
                **return** True
            **else**
                $\mathrm{numevaluations} \leftarrow \mathrm{numevaluations} - 1$
                **if** $\mathrm{numevaluations} \leq 0$ **then**
                    **return** False
        $\mathrm{features} \leftarrow \mathrm{features}[1 :]$
        $\mathrm{actions} \leftarrow \mathrm{actions}[1 :]$

        $x, y \leftarrow$ update location given action                $\triangleright$ take new action
        $\mathrm{features}[t + 1] \leftarrow$ features at location (x, y)
        $\mathrm{actions}[t] \leftarrow \mathrm{action}$

---

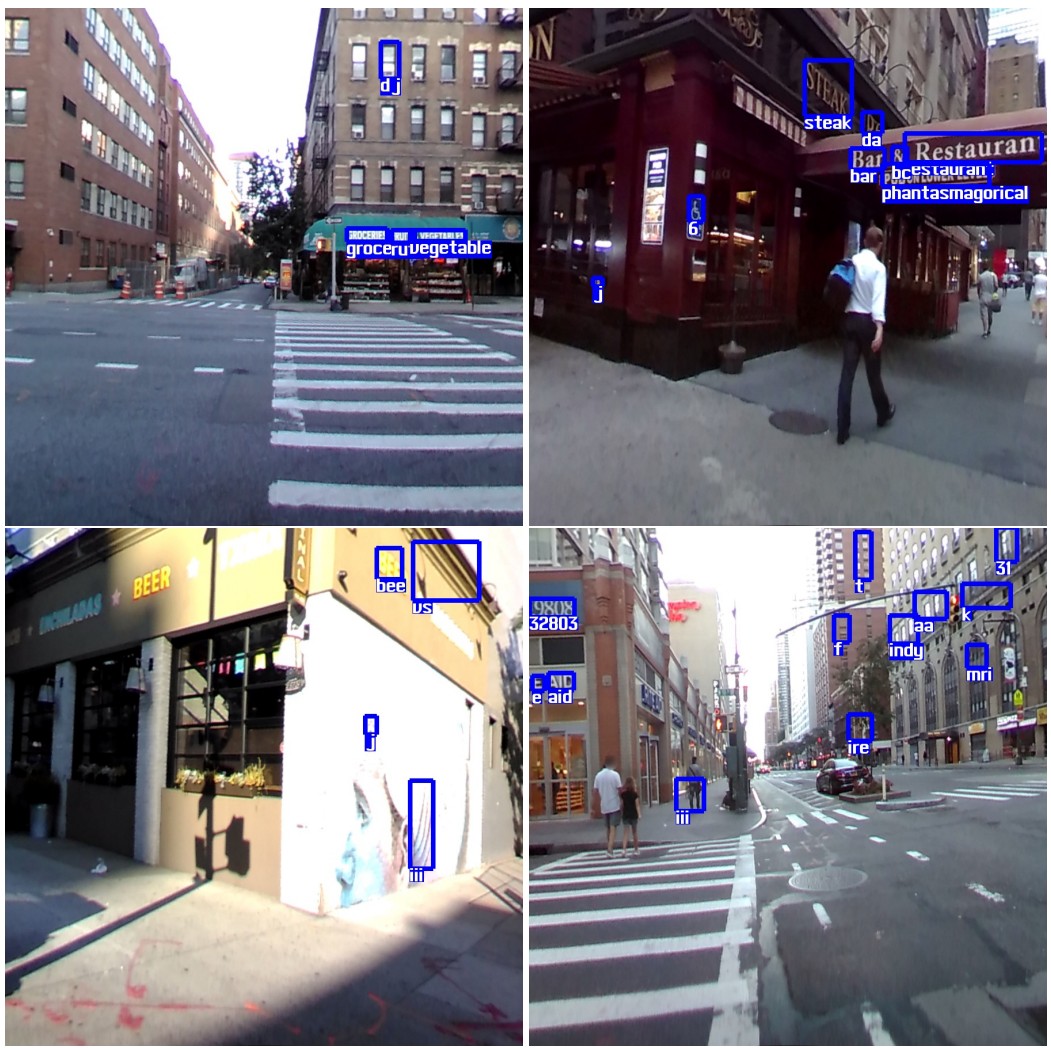

Figure 3: Result of running the text recognizer of Gupta et al. (2016) on four examples of the Hell's Kitchen neighborhood. **Top row:** two positive examples. **Bottom row:** example of false negative (left) and many false positives (right)

## 12  LANDMARK CLASSIFICATION

While the guide has access to the landmark labels, the tourist needs to recognize these landmarks from raw perceptual information. In this section, we study landmark classification as a supervised learning problem to investigate the difficulty of perceptual grounding in Talk The Walk.

The Talk The Walk dataset contains a total of 307 different landmarks divided among nine classes, see Figure 4 for how they are distributed. The class distribution is fairly imbalanced, with shops and restaurants as the most frequent landmarks and relatively few play fields and theaters. We treat landmark recognition as a multi-label classification problem as there can be multiple landmarks on a corner[5].

For the task of landmark classification, we extract the relevant views of the 360 image from which a landmark is visible. Because landmarks are labeled to be on a specific corner of an intersection, we assume that they are visible from one of the orientations facing away from the intersection. For example, for a landmark on the northwest corner of an intersection, we extract views

---

[5]Strictly speaking, this is more general than a multi-label setup because a corner might contain multiple landmarks of the same class.

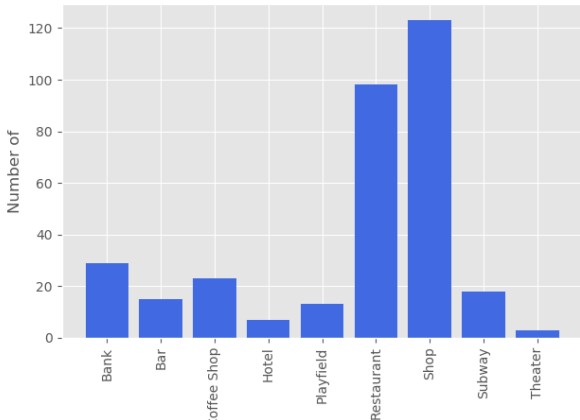

Figure 4: Frequency of landmark classes

| Features | Train loss | Valid Loss | Train F1 | Valid F1 | Valid prec. | Valid recall |
|---|---|---|---|---|---|---|
| All positive | - | - | - | 0.39313 | 0.26128 | 1 |
| Random (0.5) | - | - | - | 0.32013 | 0.24132 | 0.25773 |
| Textrecog | 0.01462 | 0.01837 | 0.31205 | 0.31684 | 0.2635 | 0.50515 |
| Fasttext | 0.00992 | 0.00994 | 0.24019 | 0.31548 | 0.26133 | 0.47423 |
| Fasttext (100 dim) | 0.00721 | 0.00863 | 0.32651 | 0.28672 | 0.24964 | 0.4433 |
| ResNet | 0.00735 | 0.00751 | 0.17085 | 0.20159 | 0.13114 | 0.58763 |
| ResNet (256 dim) | 0.0051 | 0.00748 | 0.60911 | 0.31953 | 0.27733 | 0.50515 |

Table 10: Results for **landmark classification**.

from both the north and west direction. The orientation-specific views are obtained by a planar projection of the full 360-image with a small field of view (60 degrees) to limit distortions. To cover the full field of view, we extract two images per orientation, with their horizontal focus point 30 degrees apart. Hence, we obtain eight images per 360 image with corresponding orientation $v \in \{N1, N2, E1, E2, S1, S2, W1, W2\}$.

We run the following pre-trained feature extractors over the extracted images:

**ResNet** We resize the extracted view to a 224x224 image and pass it through a ResNet-152 network He et al. (2016) to obtain a 2048-dimensional feature vector $S_{x,y,v}^{resnet} \in \mathbb{R}^{2048}$ from the penultimate layer.

**Text Recognition** We use a pre-trained text-recognition model Gupta et al. (2016) to extract a set of text messages $S_{x,y,v}^{text} = \{R_\beta^{text}\}_{\beta=0}^{B}$ from the images. Local businesses often advertise their wares through key phrases on their storefront, and understanding this text might be a good indicator of the type of landmark. In Figure 3, we show the results of running the text recognition module on a few extracted images.

For the text recognition model, we use a learned look-up table $E^{text}$ to embed the extracted text features $\mathbf{e}_{x,y,v}^\beta = E^{text}(R_\beta^{text})$, and fuse all embeddings of four images through a bag of embeddings, i.e., $\mathbf{e}^{fused} = \sum_{v \in \text{relevant views}} \sum_\beta e_{x,y,v}^\beta$. We use a linear layer followed by a sigmoid to predict the probability for each class, i.e. $\text{sigmoid}(W\mathbf{e}^{fused} + b)$. We also experiment with replacing the look-up embeddings with pre-trained FastText embeddings Bojanowski et al. (2016). For the ResNet model, we use a bag of embeddings over the four ResNet features, i.e. $\mathbf{e}^{fused} = \sum_{v \in \text{relevant views}} S_{x,y,v}^{resnet}$, before we pass it through a linear layer to predict the class probabilities: $\text{sigmoid}(W\mathbf{e}^{fused} + b)$. We also conduct experiments where we first apply PCA to the extracted ResNet and FastText features before we feed them to the model.

To account for class imbalance, we train all described models with a binary cross entropy loss weighted by the inverted class frequency. We create a 80-20 class-conditional split of the dataset into a training and validation set. We train for 100 epochs and perform early stopping on the validation loss.

The F1 scores for the described methods in Table 10. We compare to an "all positive" baseline that always predicts that the landmark class is visible and observe that all presented models struggle to outperform this baseline. Although 256-dimensional ResNet features achieve slightly better precision on the validation set, it results in much worse recall and a lower F1 score. Our results indicate that perceptual grounding is a difficult task, which easily merits a paper of its own right, and so we leave further improvements (e.g. better text recognizers) for future work.

## 13 DATASET DETAILS

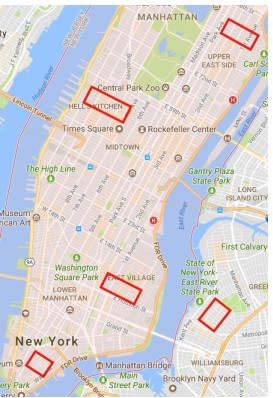

| Neighborhood | #success | #failed | #disconnects |
|---|---|---|---|
| Hell's Kitchen | 2075 | 762 | 867 |
| Williamsburg | 2077 | 683 | 780 |
| East Village | 2035 | 713 | 624 |
| Financial District | 2042 | 607 | 497 |
| Upper East | 2081 | 359 | 576 |
| Total | 10310 | 3124 | 3344 |

Figure 5: Map of New York City with red rectangles indicating the captured neighborhoods of the Talk The Walk dataset.

Table 11: Dataset statistics split by neighborhood and dialogue status.

**Dataset split** We split the full dataset by assigning entire 4x4 grids (independent of the target location) to the train, valid or test set. Specifically, we design the split such that the valid set contains at least one intersection (out of four) is not part of the train set. For the test set, all four intersections are novel. See our source code, available at URLANONYMIZED, for more details on how this split is realized.

**Example**

```
Tourist:  ACTION:TURNRIGHT ACTION:TURNRIGHT
Guide:    Hello, what are you near?
Tourist:  ACTION:TURNLEFT ACTION:TURNLEFT ACTION:TURNLEFT
Tourist:  Hello, in front of me is a Brooks Brothers
Tourist:  ACTION:TURNLEFT ACTION:FORWARD ACTION:TURNLEFT ACTION:TURNLEFT
Guide:    Is that a shop or restaurant?
Tourist:  ACTION:TURNLEFT
Tourist:  It is a clothing shop.
Tourist:  ACTION:TURNLEFT
Guide:    You need to go to the intersection in the northwest corner of the map
Tourist:  ACTION:TURNLEFT
Tourist:  There appears to be a bank behind me.
Tourist:  ACTION:TURNLEFT ACTION:TURNLEFT ACTION:TURNRIGHT ACTION:TURNRIGHT
Guide:    Ok, turn left then go straight up that road
Tourist:  ACTION:TURNLEFT ACTION:TURNLEFT ACTION:TURNLEFT ACTION:FORWARD ACTION:TURNRIGHT
          ACTION:FORWARD ACTION:FORWARD ACTION:TURNLEFT ACTION:TURNLEFT ACTION:TURNLEFT
Guide:    There should be shops on two of the corners but you
          need to go to the corner without a shop.
Tourist:  ACTION:FORWARD ACTION:FORWARD ACTION:FORWARD ACTION:TURNLEFT ACTION:TURNLEFT
Guide:    let me know when you get there.
Tourist:  on my left is Radio city Music hall
```

```
Tourist:  ACTION:TURNLEFT ACTION:FORWARD ACTION:TURNLEFT ACTION:TURNRIGHT ACTION:TURNRIGHT
Tourist:  I can't go straight any further.
Guide:    ok. turn so that the theater is on your right.
Guide:    then go straight
Tourist:  That would be going back the way I came
Guide:    yeah. I was looking at the wrong bank
Tourist:  I'll notify when I am back at the brooks brothers, and the bank.
Tourist:  ACTION:TURNRIGHT
Guide:    make a right when the bank is on your left
Tourist:  ACTION:FORWARD ACTION:FORWARD ACTION:TURNRIGHT
Tourist:  Making the right at the bank.
Tourist:  ACTION:FORWARD ACTION:FORWARD
Tourist:  I can't go that way.
Tourist:  ACTION:TURNLEFT
Tourist:  Bank is ahead of me on the right
Tourist:  ACTION:FORWARD ACTION:FORWARD ACTION:TURNLEFT
Guide:    turn around on that intersection
Tourist:  I can only go to the left or back the way I just came.
Tourist:  ACTION:TURNLEFT
Guide:    you're in the right place. do you see shops on the corners?
Guide:    If you're on the corner with the bank, cross the street
Tourist:  I'm back where I started by the shop and the bank.
Tourist:  ACTION:TURNRIGHT
Guide:    on the same side of the street?
Tourist:  crossing the street now
Tourist:  ACTION:FORWARD ACTION:FORWARD ACTION:TURNLEFT
Tourist:  there is an I love new york shop across the street on the left from me now
Tourist:  ACTION:TURNRIGHT ACTION:FORWARD
Guide:    ok. I'll see if it's right.
Guide:    EVALUATE_LOCATION
Guide:    It's not right.
Tourist:  What should I be on the look for?
Tourist:  ACTION:TURNRIGHT ACTION:TURNRIGHT ACTION:TURNRIGHT
Guide:    There should be shops on two corners but you need to be on one of the corners
          without the shop.
Guide:    Try the other corner.
Tourist:  this intersection has 2 shop corners and a bank corner
Guide:    yes. that's what I see on the map.
Tourist:  should I go to the bank corner? or one of the shop corners?
          or the blank corner (perhaps a hotel)
Tourist:  ACTION:TURNLEFT ACTION:TURNLEFT ACTION:TURNRIGHT ACTION:TURNRIGHT
Guide:    Go to the one near the hotel. The map says the hotel is a little
          further down but it might be a little off.
Tourist:  It's a big hotel it's possible.
Tourist:  ACTION:FORWARD ACTION:TURNLEFT ACTION:FORWARD ACTION:TURNRIGHT
Tourist:  I'm on the hotel corner
Guide:    EVALUATE_LOCATION
```

## 14 MECHANICAL TURK INSTRUCTIONS

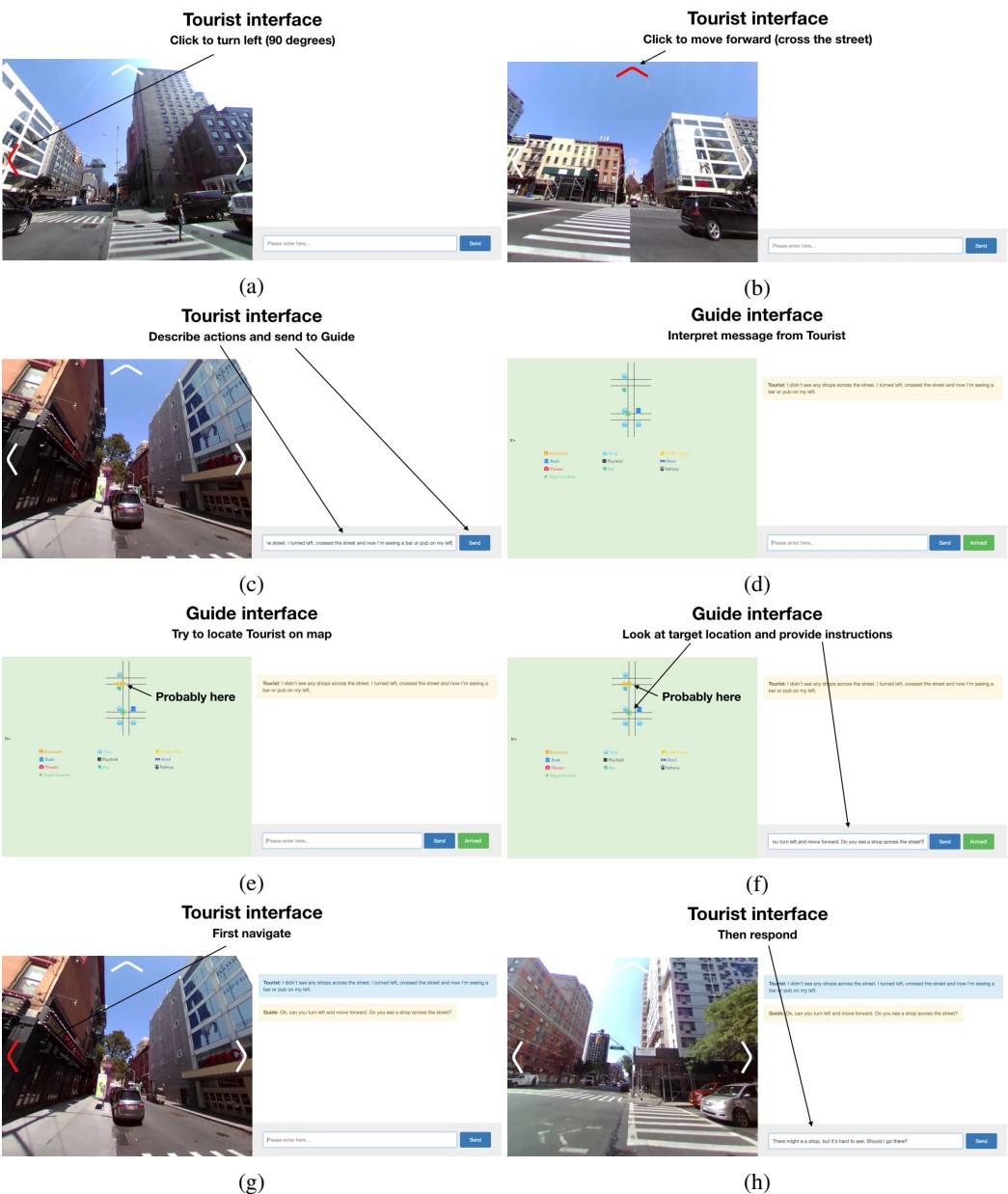

Figure 6: Set of instructions presented to turkers before starting their first task.

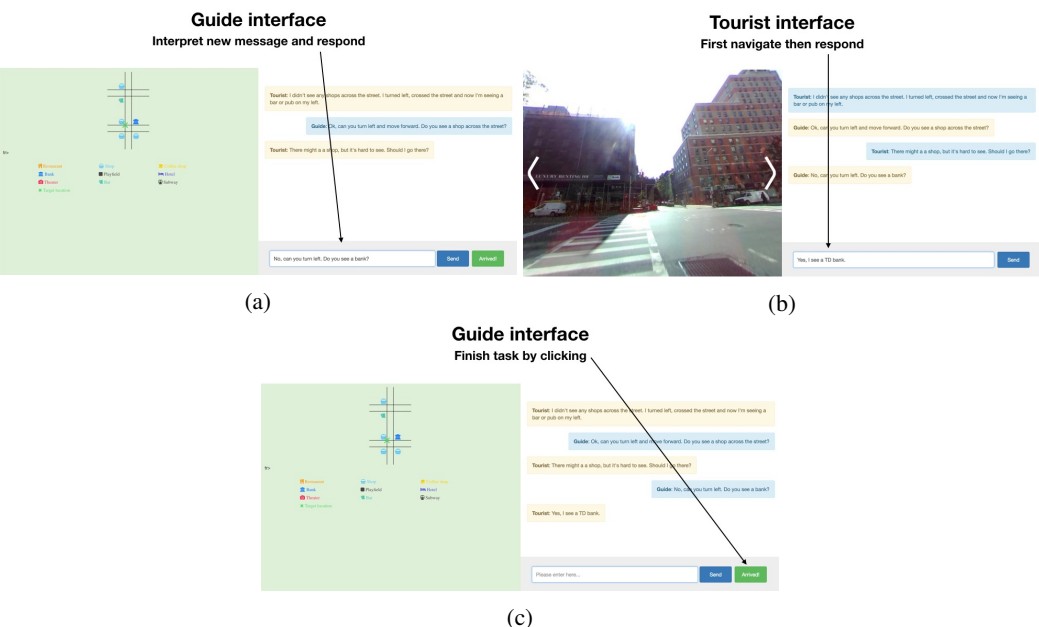

Figure 7: (cont.) Set of instructions presented to turkers before starting their first task.

