# OpenReview forum: "Talk The Walk: Navigating Grids in New York City through Grounded Dialogue"
_ICLR.cc/2019/Conference_

### Official Review · AnonReviewer3 · 2018-11-01
**Emergent Language is Easier than Natural Language**

**Rating:** 4
**Confidence:** 3

**Review:**

The primary contribution of this work is a dataset for action following through dialogue.  The authors collect a comparatively small dataset in terms of language but one which contains real images and dialogue.

There are a number of aspects of the proposed approach which I found hard to follow/justify.  First off, I was unclear on the details of the collected data (e.g. average action sequence length, dialogue length, lexical types/tokens, etc).  There's a claim of 62 acts which sums both dialogue and actions with averages of 8 and 9 dialogue acts for tourist/guide implying 45 move actions?  on a 4x4 grid?  Is it safe therefore to assume that the example dialogue is therefore atypical? It's very hard to figure out based on the number of steps across the different tables what the model should be aiming for.  Also, in 2.3 does the claim that they "successfully complete the task" mean in the 76.74% of cases where they succeed or did they succeed in 100% of cases and then a new human eval was run afterwards which performed worse?

The primary modeling result appears to be the success of emergent language and the bold claim that humans are bad at localizing.  This doesn't feel intuitively true from the example dialogue, but the NLG system samples to appear to be quite bad which makes me worried that it's not so much that humans are bad localizers but that the model's NLU/NLG system is quite weak and maybe there's a problem with the data-collection procedure.  Additional justification and analysis would be appreciated.

As I understand the paper right now:
1. Humans talking to one another do very well on the task and achieve success very quickly.
2. Emergent language can do better at the task though their approach is very sub-optimal (requiring 2-3x the number of steps).
3. The currently proposed NLU/NLG mechanisms are very weak and cannot produce or correctly interpret actual language.

There are many moving pieces in this paper (e.g. extracting text from images vs detections), there doesn't seem to be any pretraining of the decoder, etc which makes it very hard for me to understand what's going wrong.  The results in this paper, don't convince me that emergent language is better than natural language or that agents are better communicators than humans, but that the data-collection methodology was faulty leading to lots of failures.

I haven't touched on the MASC aspect and how this compares to existing work on interpretable spatial relations and questions as to why various architectural choices were made though the paper would obviously benefit from that discussion as well.

I found this paper very confusing to read.  It relies heavily on 11 pages of appendices (where it puts all of the related work) and still fails to clearly explain its contributions or justify its claims.

Minor: URLs intermittently anonymized page 12 vs 19

---

> ### Author Response · Authors · 2018-11-25
> **Improved paper presentation to resolve misunderstanding of our experimental findings**
>
> Thank you for taking the time to review our manuscript :)
>
> We believe there is a misinterpretation of our experimental findings, most likely due to the lack of clarity in the presentation of our results. We did not intend to communicate that the main takeaways are 1) success of emergent language and 2) the bold claim that humans are bad at localizing. Concerning 1), we have included emergent localization to put our natural language experiments in perspective, as well as to show that the full task is do-able under some simplifications. For instance, it points out that our conditional (natural) language models do not achieve high localization accuracy because of the guide model not being able to handle messages that are grounded in multiple observations and actions (because it can do so for emergent communication). Concerning 2), we wanted to convey that the human baseline does not rely as much on the localization capabilities as our constructed random walk protocol. This is partly supported by low localization performance from extracted natural language utterances, although we agree that this might be due to modeling issues. We removed this paragraph from the results section and rewrote this entire section to better highlight our main findings.
>
> We also want to reiterate that the main modeling contributions are:
> - We establish an initial baseline on the full navigation task, by designing a random walk protocol that utilizes tourist and guide agents trained for localization.
> - We show that the localization task is challenging, as it requires communication of a short random path (i.e. observations and actions from multiple time steps)
> - For emergent communication, we show that the MASC mechanism is a crucial component to successfully learn a protocol that grounds observations and actions over multiple time steps
> - Localization performance from human utterances is much worse than that from emergent communication
> - Our analysis shows that tourist generation models can only produce natural language utterances grounded in a single landmark observation and are less successful when conditioned on observations and actions of multiple time steps.
>
> We have rewritten the paper from Section 3 onwards to improve the clarity of our key findings. Specifically, we have made the following revisions:
> - We start Section 3 by explaining that our focus is on establishing baselines for the full task, subsequently describing the random walk protocol, and then moving on to the localization task.
> - After we introduce the simplifying assumptions for the localization task, we explain in one paragraph what the remaining challenges of the task are.
> - At the beginning of Section 5, we now briefly summarize our main findings.
> - We divided the findings of different experiments in separate subsections: analysis of localization task, emergent localization, natural language localization, and localization-based baselines.
> - We refactored the result subsection on natural language localization:
> i) removed the claim that humans are bad localizers,
> ii) added a paragraph explaining that generated utterances from a tourist model achieve better localization accuracy than extracted human utterances, as well as added an explanation that this is due improved grounding on a single observation (see point below),
> iii) added that tourist generation models can not (yet) produce utterances about multiple observations
> - We fixed the caption of Table 8 and moved the examples to the main paper as it illustrates well why the generated tourist utterances lead to better localization accuracy than human utterances from the dataset.
>
> Q: Clarifications regarding dataset statistics
> A:  In section 2.3, the phrase "before they successfully complete the task" indicates that the average number of acts are calculated over the successfully completed dialogues in the collected dataset (i.e. 76.74% of all dialogues). We use the calculated task success rate (76.74%) as human accuracy., i.e. this statistic is not gathered by a separate human evaluation. We have updated section 2.3 to point this out.
>
> Q: Isn't 45 actions a lot for 4x4 grids?
> A: Yes, but most of these actions are turning actions (30 turn lefts/rights), which do not move the tourist to new x,y coordinates.

---

### Official Review · AnonReviewer1 · 2018-11-03
**A challenging new task and dataset**

**Rating:** 7
**Confidence:** 4

**Review:**

The paper introduces a new task called "Talk the Walk", where a tourist and a guide has to communicate in natural language to reach a common goal. It also introduces strong baselines for the task. The descriptions are thorough and clear. My only worry is that the task is too hard and has too many complexities to be a stand alone task.  Future work will probably focus on sub-parts of the task.

---

> ### Author Response · Authors · 2018-11-25
> **Thank you for the positive feedback; can you please elaborate?**
>
> Thanks for your positive feedback on our work. It would be really helpful if you could elaborate on the usefulness and challenges of the new task, the introduced baselines, or the descriptions of our experiments.

---

### Official Review · AnonReviewer2 · 2018-11-03
**A realistic dataset on dialogs for navigation, with a report of some early studies**

**Rating:** 6
**Confidence:** 4

**Review:**

The paper introduces a new dataset "Talk the Walk" that are dialogs
between a guide and a tourist, where the guide is to help the tourist
navigate to a target location.  The guide has access to a map and the
target location, but he relies on the tourist to communicate her
state (location) by natural language.

Pros.:

The task represented in the dataset can be highly challenging, and
respectable effort went into creating the dataset based on real city
neighborhoods.   The description and analysis of the dataset are
detailed.  The paper is well written up to the end of page 3.

The dataset by itself is a good contribution to the scientific
community when it is shared.  There could be many topics open
for studying within the same data.

The list of references and related work is exceptionally thorough
and useful for researchers interested in the topic.

Cons.:

The description of the experiments done with the dataset, however,
suffers from being overly cryptic.  The methods are not sufficiently
motivated, very few alternatives are presented and argued against,
and the several sections give a dry report of the sequences of things
the authors did.  It is not clear how others may find value in the
results and conclusions.

While the paper opens with the emphasis of a real-world setting,
after a series of simplifications (e.g. landmark typing, perfect perception)
it seems that much of the full complexity of the natural task is taken out,
and the main goal of the study is no longer clear.
For example, since there are only 9 types of landmarks,
in a small neighborhood there are not many combinations to draw reference to.
Simple observation sequences of such can easily narrow down the location
uncertainty.  It is important to highlight what the remaining open
issues are that make the task still challenging.

Misc.:

Examples are missing in the discussion of the experimental tasks.
e.g. in the study of emergent language, what could be a message that a
tourist may generate to describe his location?  what makes it hard for
the guide to decode it?  Likewise, what could be an example state of
the tourist and the description of that state in natural language?
Without the examples, it is difficult for the reader to have a sense
of the challenges in each task.

Table 8 is the first place where (finally) some utterances are presented.
However the description in the table or in the text is not sufficient
to convey the point that is supposed to be explained by the example.

The descriptions of the landmarks are restricted to the type of business
at the location with 9 possibilities.  Is the list of 9 exhaustive?
Are there any exceptions  (e.g. schools)?  How are such exceptions
represented and treated in the dialogs?

To what extent the difficulty of the tasks depends on the variability
of the combination of landmarks visible at each location?

What could be a simplest way to do this without neural-modeling?
e.g. with the many limitations that are built into the task and its
representations, will a simple decision tree based instruction method suffice?
Or a traditional algorithm that relies on repeated exploration and evaluation?
It is surprising that such possibilities are not even mentioned.
A complex neural architecture does not seem to be well justified unless
it is motivated by the need to overcome limitations of a classical method.

Irrelevant to the research effort, a thought about the dataset is that,
in these days with popular uses of GPS, the reliance on such dialogs for
navigation feels a bit backwards.

---

> ### Author Response · Authors · 2018-11-25
> **Substantially revised paper to better motivate experiments**
>
> Thank you for your extensive review of our work! We find it encouraging that you describe the dataset as a useful contribution to the community and the related work section exceptionally thorough and useful for researchers interested in this topic.
>
> We apologize for the lack of clarity in the experimental section, and we agree that this part of the paper did not communicate our motivations and findings very well. The main focus of the current work is to establish a minimal baseline for the full navigation task, so as to facilitate future research on the task. In order to do so, we design a random walk protocol that utilizes tourist and guide agents trained for localization. This simple baseline only relies on a localization model; hence we primarily focus on this sub-task. Even though our objective is to accomplish this task via natural language communication, we also investigate emergent communication baselines in order to provide useful comparisons and to get a better understanding of the difficulty of the task.
>
> Concerning the challenges of the localization task after the introduced simplifications, we investigate this question in the first two paragraphs of section 5 (5.1 in the revised paper). The task requires communication of a short path---i.e., not only a sequence of landmark observations but also actions---to achieve high localization accuracy. That is, the guide needs to decode observations from multiple time steps, as well as understand their 2D spatial arrangement as communicated via the sequence of actions. For emergent communication, we show that the introduced MASC mechanism is a crucial component to achieve high accuracy on the localization task, by successfully learning a protocol that can ground such actions in the guide's overhead map.
>
> We have rewritten the paper from Section 3 onwards in order to present the above points better. Specifically, we have made the following changes:
> - We start Section 3 by explaining that our focus is on establishing baselines for the full task, subsequently describing the random walk protocol, and then moving on to the localization task.
> - After we introduce the simplifying assumptions for the localization task, we explain in one paragraph what the remaining challenges of the task are.
> - At the beginning of Section 5, we now briefly summarize our main findings.
> - We divided the findings of different experiments in separate subsections: analysis of localization task, emergent localization, natural language localization, and localization-based baselines.
> - We refactored the result subsection on natural language localization:
> 1) removed the claim that humans are bad localizers,
> 2) added a paragraph explaining that generated utterances from a tourist model achieve better localization accuracy than extracted human utterances, as well as added an explanation that this is due to improved grounding on a single observation (see point below),
> 3) added that tourist generation models can not (yet) produce utterances about multiple observations
> - We fixed the caption of Table 8 and moved the examples to the main paper as it illustrates well why the generated tourist utterances lead to better localization accuracy than human utterances from the dataset.
>
> These modifications should help clarify our experiments, motivations, and findings. Below, we will address some specific concerns and questions.
> Q: Is the list of landmarks exhaustive?
> A: Yes, the list of 9 landmarks is exhaustive, and not-annotated buildings (e.g. schools) are not shown on the map. As a result, the human annotators might sometimes talk about whether a particular observation is on the overhead map or not.
>
> Q: To what extent does the difficulty of the tasks depends on the variability
> of the combination of landmarks visible at each location?
> A: This is a great question - if the landmark observation is unique (i.e. the only one on the map), we have absolute certainty about the location of the tourist. However, the overhead maps have coarsely annotated landmarks so that the landmarks are often visible at multiple locations. We investigate the difficulty of the localization task (on average) in the first two paragraphs of section 5 (5.1 in the revised version) by analyzing the upper bound performance on the localization task when varying the length of a random path on the map. We find that communicating a single observation is not sufficient; instead, a random path (of length T >= 2) needs to be communicated to achieve high localization accuracy (>=86%).

---

### Public Comment · (anonymous) · 2018-10-03
**Great task but is the Perfect Perception oversimplified the problem?**

As mentioned in the paper, "To ensure that perception is not the limiting factor when investigating localization capabilities of models, we assume “perfect perception”: in lieu of the 360 image view, the tourist is given the landmarks at its current location," does this mean that the tourist does not need to learn the grounding between language and vision? In this way, doesn't it make the task too easy (even surpass human) because the tourist just need to say every landmarks provided one-by-one instead of any other real conversation?In the human evaluation, did they also get all the landmarks?  And once the guides get all the names of the landmarks, it is also easy for them to find the location by matching the words (or corresponding vectors).

Can some examples of your model's natural language generation text be provided? That will be helpful :)

---

> ### Author Response · Authors · 2018-10-04
> **Perfect Perception is to establish sensible baselines, but is not part of the full task**
>
> Thank you for your interest in our work!
>
> The perfect perception assumption is not part of the full task that we propose (i.e. the gathered human data is using 360-degree navigation with two-way natural language communication). We simplified the perception aspect in our experiments because initial findings suggested that perceptual grounding is difficult for simple baseline models (see Section 12 of the Appendix). This indeed makes it easier for the tourist to produce a message about the observed landmarks. Nevertheless, the localization task is still challenging because only communicating the list of observed landmarks is not sufficient for accurate localization (see Table 2:  the upper bound on localization performance is ~56% for taking T=3 actions). In other words, the tourist also needs to communicate their actions to the guide, who is responsible for grounding the entire trajectory (landmarks + actions) in the overhead map before predicting the location. We believe that this sub-task is non-trivial and develop a novel action-grounding mechanism (MASC) that is essential to obtain high localization accuracy with emergent language, as well as improves performance for natural language.
>
> You can find samples of the natural language model (for different decoding strategies) in Table 8 of the Appendix :)

---

### Meta-Review · Area_Chair1 · 2018-11-07
**Great dataset, but many questions remain open**

**Confidence:** 4
**Recommendation:** Reject

**Metareview:**

This paper introduces a newly collected dataset of natural language interactions between a tourist and a guide for localization and navigation.  The paper also includes baseline experiments with a reasonably novel approach.
The task is well motivated (although an open question remains due to GPS, comment by reviewer 1), but the description of the dataset and collection, approach and experiments were not ideal in the first version of the paper. Much of the information was pushed to the appendix and it was hard to follow the paper without going back and forth, and even then some points were missing. Authors rewrote parts of the paper to address these concerns, but there are still some open questions. For example, is it possible to have sub-tasks, given the task is complex and may not be easy to accomplish as a whole? Or could simple LSTM be another baseline (the final review of the third reviewer)?